# Respiratory Health and Urinary Trace Metals among Artisanal Stone-Crushers: A Cross-Sectional Study in Lubumbashi, DR Congo

**DOI:** 10.3390/ijerph17249384

**Published:** 2020-12-15

**Authors:** Tony Kayembe-Kitenge, Irene Kabange Umba, Paul Musa Obadia, Sebastien Mbuyi-Musanzayi, Patient Nkulu Banza, Patrick D. M. C. Katoto, Cyrille Katshiez Nawej, Georges Kalenga Ilunga, Vincent Haufroid, Célestin Banza Lubaba Nkulu, Tim Nawrot, Benoit Nemery

**Affiliations:** 1Faculty of Medicine, University of Lubumbashi, Lubumbashi, Democratic Republic of the Congo; tonykayemb@gmail.com (T.K.-K.); kabangeirene6@gmail.com (I.K.U.); musa.p.obadia@gmail.com (P.M.O.); sebambuyi@gmail.com (S.M.-M.); patientnkulu@gmail.com (P.N.B.); ilukalenga@gmail.com (G.K.I.); clubabankulu2017@gmail.com (C.B.L.N.); 2Department of Internal Medicine, Faculty of Medicine, University of Malemba-Nkulu, Malemba, Democratic Republic of the Congo; 3Centre for Environment and Health, Department of Public Health and Primary Care, KU Leuven, 3000 Leuven, Belgium; patrick.katoto@kuleuven.be (P.D.M.C.K.); tim.nawrot@kuleuven.be (T.N.); 4Department of Internal Medicine and Prof Lurhuma Biomedical Research Laboratory, Mycobacterium Unit, Faculty of Medicine, Catholic University of Bukavu, Bukavu, Democratic Republic of the Congo; 5Department of Internal Medicine, Faculty of Medicine, University of Kolwezi, Kolwezi, Democratic Republic of the Congo; katshiez@yahoo.fr; 6Louvain Center for Toxicology and Applied Pharmacology, Université Catholique de Louvain, 1200 Brussels, Belgium; vincent.haufroid@uclouvain.be

**Keywords:** respiratory health, spirometry, metal biomonitoring, mineral dust, mining, Lubumbashi

## Abstract

Background: Thousands of artisanal workers are exposed to mineral dusts from various origins in the African Copperbelt. We determined the prevalence of respiratory symptoms, pulmonary function, and urinary metals among artisanal stone-crushers in Lubumbashi. Methods: We conducted a cross-sectional study of 48 male artisanal stone-crushers and 50 male taxi-drivers using a standardized questionnaire and spirometry. Concentrations of trace metals were measured by Inductively Coupled - Plasma Mass Spectrometry (ICP-MS) in urine spot samples. Results: Urinary Co, Ni, As, and Se were higher in stone-crushers than in control participants. Wheezing was more prevalent (*p* = 0.021) among stone-crushers (23%) than among taxi-drivers (6%). In multiple logistic regression analysis, the job of a stone-crusher was associated to wheezing (adjusted Odds Ratio 4.45, 95% Confidence Interval 1.09–18.24). Stone-crushers had higher values (% predicted) than taxi-drivers for Forced Vital Capacity (105.4 ± 15.9 vs. 92.2 ± 17.8, *p* = 0.048), Forced Expiratory Volume in 1 Second (104.4 ± 13.7 vs. 88.0 ± 19.6, *p* = 0.052), and Maximum Expiratory Flow at 25% of the Forced Vital Capacity (79.0.1 ± 20.7 vs. 55.7 ± 30.1, *p* = 0.078). Conclusion: Stone-crushers were more heavily exposed to mineral dust and various trace elements than taxi-drivers, and they had a fourfold increased risk of reporting wheezing, but they did not have evidence of more respiratory impairment than taxi-drivers.

## 1. Introduction

Several studies have documented high to very high levels of mineral dust in worksites—and also the surrounding environment—where stones or rocks are crushed or milled using various types of mechanical crushers to produce aggregates for use in the construction of roads and buildings [1,2,3,4,5,6]. In these publications, the main emphasis was on the health risks associated with high exposures to free crystalline silica, i.e., mainly quartz, the content of which depends on the nature of the materials that are being crushed. Chronic inhalation of free crystalline silica may lead, usually after several years of exposure, to silicosis, chronic obstructive pulmonary disease (COPD), and lung cancer, and contribute to pulmonary tuberculosis and autoimmune diseases, such as systemic sclerosis [7]. Moreover, chronic inhalation of other poorly soluble low-toxicity particles (“biopersistent granular dust”) is also associated with the development of COPD [8].

Lubumbashi is the second largest city of the Democratic Republic of Congo (DRC) and the capital of the Haut-Katanga Province, which is situated in the African Copperbelt, an area of intense past and current mining and processing of copper and cobalt ores. Artisanal mining has also become widespread in the past 20 years, with tens of thousands of young people involved. These activities have led to widespread environmental pollution by various trace metals [9]. Biomonitoring studies have documented that people living close to mining activities have high internal exposure to cobalt and other trace metals [10,11]. Besides copper and cobalt mining and smelting, many other industrial and artisanal activities take place in the area. One of these artisanal activities consists of crushing stones to produce gravel for use in the construction of buildings and roads. In Lubumbashi, hundreds of poor people are engaged in artisanal stone crushing using hand tools (see photographs in Figure 1). The degree of exposure to toxic metals and the respiratory impact of this dusty work have not been studied in this population.

We, therefore, performed a cross-sectional study using urinary biomonitoring to assess metal exposure and spirometry to assess pulmonary function among artisanal stone crushers, taking drivers of collective taxis as controls without occupational exposure to mineral dust.

## 2. Methods

This cross-sectional study took place in Lubumbashi in October 2014 (dry season). Potential participants were recruited by convenience sampling at their place of work over a period of 6 days, during which 48 men working as artisanal stone-crushers and 50 men working as drivers of collective minibus taxis were included. All participants found at their workplace were invited to participate, and those giving their oral consent replied to a respiratory questionnaire, performed spirometry, and provided a spot sample of urine, all procedures being done at their worksite. The study protocol, including the oral consent procedure, was approved by the medical ethics committee of the University of Lubumbashi.

We used a combination of questionnaires, as in a study of workers performed in Algeria [12], namely the International Union Against Tuberculosis and Respiratory Diseases (IUATLD) Bronchial Symptoms Questionnaire [13], to obtain information about respiratory symptoms in the past 12 months, and the questionnaire on allergic rhinitis [14], with some additional questions related to the local context. The questions were administered face-to-face in Swahili (own translation) by the same interviewers for stone-crushers and taxi-drivers.

We performed spirometry in 68 participants (34 stone-crushers and 34 taxi-drivers) using the portable EasyOne^®^ Air device (ndd Medical Technologies, Zurich, Switzerland). In accordance with ATS/ERS (American Thoracic Society/European Respiratory Society) guidelines [15], a minimum of 3 and a maximum of 8 satisfactory forced expiration maneuvers were performed, in the sitting position and without a nose clip, and the highest values for the Forced Vital Capacity (FVC), Forced Expiratory Volume in 1 Second (FEV1), Peak Expiratory Flow (PEF), Maximal Mid-Expiratory Flow (MEF25-75), Maximal Expiratory Flow at 50% (MEF50), and 25% (MEF25) of the FVC obtained from the best curves, were retained. However, because the curves were not displayed on a computer screen during the forced expiration maneuvers, an experienced lung function expert (Geert Celis, Pulmonary Function Laboratory, UZ Leuven) later independently checked the quality of the printed spirometry curves and scored them as follows: Score 0: Unacceptable; score 1: FEV1 (and PEF) probably reliable, but FVC not acceptable; score 2: FVC probably reliable, but FEV1 not acceptable; score 3: Both FEV1 and FVC acceptable. Only spirometries with a score of 3 were used for assessing FEV1/FVC, MEF25-75, MEF50, and MEF25. Height was measured using a measuring rod. Percent predicted values for FEV1 and FVC were obtained for subjects of African descent, as provided by the EasyOne software.

We obtained a spot sample of urine from 75 participants (41 stone-crushers and 34 taxi-drivers), who were instructed to void urine, after hand-washing and without contaminating the sample by their hands, into a 40 mL polystyrene vial with screw cap (Plastic-Gosselin, Hazebrouck, France). Urine was transferred the same day into cryovials, which were kept frozen and later shipped in cool-boxes to Belgium by commercial flights. The concentrations of 24 elements [Lithium (Li), Beryllium (Be), Aluminium (Al), Vanadium (V), Chromium (Cr), Manganese (Mn), Cobalt (Co), Nickel (Ni), Copper (Cu), Zinc (Zn), Arsenic (As), Selenium (Se), Molybdenum (Mo), Cadmium (Cd), Indium (In), Tin (Sn), Antimony (Sb), Tellurium (Te), Barium (Ba), Platinum (Pt), Thallium (Tl), Lead (Pb), Bismuth (Bi) and Uranium (U)] were analyzed in 100 µL urine by Inductively Coupled Plasma-Mass Spectrometry (ICP-MS), using an Agilent 7500ce instrument (Agilent Technologies, Santa Clara, CA, USA), in the internationally accredited Laboratory of the Louvain Center for Toxicology and Applied Pharmacology (Université catholique de Louvain, Belgium) using validated methods, as previously described [16]. In brief, urine specimens were diluted quantitatively (1 + 9) with a HNO3 1%, HCl 0.5% solution containing Sc, Ge, Rh, and Ir as internal standards. Sb, Al, Cd, Pb, Mo, Te, Sn, and U were analyzed using no-gas mode, while helium mode was selected to quantify As, Cu, Co, Cr, Mn, Ni, Se, V, and Zn. Using this method, the laboratory obtained successful results in external quality assessment schemes organized by the Institute for Occupational, Environmental and Social Medicine of the University of Erlangen, Germany (G-EQUAS program) and by the Institut National de Santé Publique, Quebec.

Metal concentrations were corrected for dilution by the concentration of creatinine, as measured by using a Beckman Synchron LX 20 analyzer (Beckman Coulter GmbH, Krefeld, Germany).

For the statistical analysis, independent variables were age, height, educational level (below vs. above 12 years of study), tobacco smoking (no smoking vs. current smoker or any regular smoker at home), alcohol consumption (none vs. current drinker), living close to mining, i.e., within visible distance from home (yes vs. no), type of drinking water (always drinking water from own well vs. no), the use of personal protective mask, gloves, and glasses (yes vs. no).

The outcome variables were positive/negative replies to 4 questions on respiratory symptoms (wheezing, cough, phlegm, shortness of breath) currently or during the last 12 months, and on 5 oculo-nasal symptoms (itchy nose, itchy eyes, runny nose, sneezing, stuffy nose).

Descriptive statistics consisted of frequency and percentages for categorical variables and means with standard deviation (SD) and range for continuous variables, or geometric means with their 95% confident interval (CI) for trace metal concentrations. To compare groups, the Fisher exact test, *t*-test, and/or Mann–Whitney rank-sum tests were used. Unadjusted (uOR) and adjusted odds ratios (aOR) and associated 95% confidence intervals (CIs) were calculated to summarize the strength of association between baseline characteristics and symptoms (respiratory and nasal) among groups. A stepwise logistic regression was performed, adjusting for variables considered clinically or epidemiologically relevant, and the variables with *p*-value less than 0.2 in the bivariate analysis. The threshold level for significance was set at *p* ˂ 0.05.

Graphpad 6 (Graphpad corps, La Jolla, CA, USA, 2015) was used to perform descriptive statistics and bivariate comparisons, and JMP Pro 14.2.0 (SAS Institute Inc. Cary, NC, USA, 2019) was used for multivariate analyses.

## 3. Results

### 3.1. Study Population

The characteristics of the 98 participants are presented in Table 1. The stone-crushers were, on average, 3 years younger than the taxi-drivers (27 ± 5 vs. 30 ± 8 years, respectively), they were much less educated than the taxi-drivers (2% vs. 52% with more than 12 years education, respectively), and they were also more likely to drink exclusively water from their own wells (50% vs. 26%, respectively). The prevalence of tobacco smoking (13%) and the reported number of cigarettes smoked daily by smokers (median 5, range 1–25) were low and similar in both groups. Very few stone-crushers reported wearing protective equipment (face-mask by three subjects, gloves by two, goggles by none). All consenting subjects replied to the questionnaire, but urine sampling and spirometry could not be performed in all participants for logistic reasons, i.e., not because of refusals.

### 3.2. Urinary Biomonitoring

The urinary concentrations of trace metals/metalloids (expressed as µg/g creatinine) are presented in Table 2. For comparison, this table also shows data measured in the same laboratory using the same methods and derived from two other studies [10,16]. One study, by Hoet et al. [16], provided general reference values based on 1022 healthy adult male and female persons, all living in Belgium and having no occupational exposure to metals. The other study, by Banza et al. [10] was done among residents living in several locations in the same region as the current study, and we considered the data obtained in 179 male and female children and adults who were living at a distance of less than 3 km from a mine or metal-processing plant, to be suitable local reference values.

In general, most urines proved to be highly concentrated since the creatinine concentration averaged 2.46 g/L (SD 1.04) for the 75 participants, with no sample having a creatinine concentration below 0.7 g/L and 20 samples (10 in each group) having a concentration exceeding 3 g/L. However, the concentrations of creatinine did not differ between stone crushers and taxi-drivers.

The concentrations of five elements (Be, V, In, Pt, Bi) were below detection limits in most subjects and are, therefore, not reported. Of the 19 other measured elements, four (Co, Ni, As, and Se) exhibited significantly higher concentrations in urine from stone-crushers than in the urine of taxi-drivers, the highest contrasts being observed for Ni and As, for which the concentrations were about two-fold higher among stone-crushers than among taxi-drivers. For two elements (Sn and Sb), urinary concentrations were higher in taxi-drivers than in stone-crushers.

Because it has been recommended to exclude too concentrated samples [17], we repeated the analyses after the exclusion of samples with creatinine concentrations above 3 g/L. This did not substantially modify the results (not shown).

### 3.3. Symptoms

The proportion of participants without reported respiratory symptoms tended (*p* = 0.07) to be lower among the stone-crushers (35%) than among the taxi drivers (54%) (Table 3). However, only wheezing was significantly more prevalent among stone-crushers (23%) than among taxi-drivers (6%, *p* = 0.021) (Table 3).

In a multivariate analysis involving the entire population (Table 4), tobacco smoking and residential proximity to mining did not affect the prevalence of reported symptoms after adjustments for the various relevant variables. However, wheezing (aOR 1.17, 95% CI 1.02–1.35), cough (aOR 1.14, 95% CI 1.01–1.29), and shortness of breath (aOR 1.28, 95% CI 1.06–1.55) were more likely with increasing age (values of aOR are for a one-year increase in age). The excess of reported wheezing among stone crushers was significant (aOR 4.45, 95% 1.09–18.24) after adjustment for age, tobacco smoking, and proximity to mines.

### 3.4. Spirometry

Of the 68 participants who performed spirometry, 39 did not provide reliable spirometric results based on the quality check of the spirograms (score 0), and only 21 participants provided fully acceptable curves (score 3); three additional subjects had acceptable FEV1 (score 1) and five additional subjects had acceptable FVC (score 2). The age of the 21 participants with fully acceptable spirometry (29 ± 5 y) differed significantly (*p* = 0.02) from that of the 47 participants who failed to produce satisfactory spirometries (26 ± 6 y), but it did not differ (*p* = 0.16) from that of the 30 participants who did not perform spirometry (31 ± 1 y). Although none of the 21 participants with acceptable spirometries reported wheezing, subjects who reported nasal or other respiratory symptoms were not more likely to have failed spirometry (not shown).

Overall, the stone-crushers had better pulmonary function than the taxi-drivers, this being significant for FVC and nearly significant for FEV1 and MEF25 (Table 5). Among the 21 participants with fully acceptable spirometry, those who were free of respiratory symptoms (n = 11, 4 taxi-drivers and 7 stone-crushers) did not have better pulmonary function than those who reported at least one respiratory symptom (n = 10, 7 taxi-drivers and 3 stone-crushers), except for FVC, which was higher among the symptom-free participants (105.1 ± 4.8 vs. 91.3 ± 5.6, *p* = 0.038).

## 4. Discussion

In this cross-sectional design, we studied trace metal exposure and respiratory health of a group of male artisanal workers involved in stone-crushing in Lubumbashi, a city in the African Copperbelt well-known for its mining-related environmental pollution [18]. Compared with a control group of taxi-drivers, the stone-crushers exhibited higher urinary levels of several trace elements [Co, Ni, As and Se], and they were more likely to report wheezing. Nevertheless, the stone-crushers tended to have better pulmonary function values than the taxi-drivers.

In both groups of participants, trace metals in urine were substantially higher than reference values from industrially developed countries [16], as well as values obtained in Kinshasa, the capital city of the DR Congo, situated at more than 1000 km distance from the mining region [19]. This is consistent with a previous study [10], in which we also found high urinary levels of cobalt and other metals among the general population living in Lubumbashi as compared to international standards. However, the urinary concentrations of As and Se were higher in the stone-crushers than those found in the residents living close to mines [10], this being possibly due to their exposure to dust produced from the crushed stones. Unfortunately, we do not have information on the elemental composition of the dust produced by crushing the stones.

Another possible explanation for the high urinary As in the stone-crushers may be the presence of high concentrations of As in their drinking water since more than 50% of the stone-crushers reported drinking only well water, and well water can be highly polluted by metal(oid)s in Lubumbashi [9]. However, we found no difference in urinary As between those reporting drinking water exclusively from their own wells and those who did not (not shown). The sources of As and other trace elements among the stone crushers, therefore, requires further investigation. Recently, we have also found high metal concentrations in surface dust obtained from households in Lubumbashi [20].

On the other hand, the metal concentrations found in the present study were generally considerably lower, especially for Co and U, than those observed in artisanal miners of cobalt in Kolwezi [11] and elsewhere in the region [21], except for Pb and Mn that were relatively high in Lubumbashi. We have no explanation for the finding that our stone-crushers and taxi-drivers had higher urinary Mn concentrations than the inhabitants and diggers of Kolwezi. However, it is known that Mn can accompany copper extracted in Lubumbashi. Similarly, high values of Pb, the origin of which is unclear, have been previously found in pregnant women in Lubumbashi [22].

Regarding respiratory symptoms, logistic regression analysis revealed that wheezing, cough, and shortness of breath were positively associated with age, even in our relatively young population. However, the only significant difference found between stone-crushers and taxi-drivers concerned wheezing, which was more frequently reported by stone-crushers. The difference was significant both in bivariate analysis and in logistic regression analysis, which revealed an adjusted OR of more than 4 for wheezing among the stone-crushers compared to taxi-drivers. Within the limits of a questionnaire-based cross-sectional survey, we attribute this excess risk of wheezing to their exposure to high levels of dust. Further studies are needed to determine whether this reported wheezing corresponds to asthma.

We did not find an effect of smoking on respiratory symptoms or pulmonary function in our study. We speculate that this can be explained by the low prevalence and low daily consumption of cigarettes in our (relatively young) study population.

Against our expectation, pulmonary function was or tended to be better among stone-crushers than among taxi-drivers, at least as assessed among the minority of participants in whom satisfactory spirometries could be obtained. The problem of obtaining good measurements of pulmonary function is not often acknowledged in published surveys done in low-income countries, where precarious field conditions, including unavailable electric power supply, render spirometry testing much less reliable than when spirometry is done in well-equipped pulmonary function laboratories. Although the ndd EasyOne portable spirometer has been recommended for epidemiological studies (e.g., in the BOLD studies) [23], its use “in the field” without a link to a computer allowing a visual control of the maneuver during the performance of the test, renders quality control almost impossible by the operator. This explains why a later check of the printed spirometry tracings led to excluding the tests of more than half of our participants, which is in accordance with a study done in Ghana among gold miners and farmers [24]. Nevertheless, if we assume that the available satisfactory measurements of pulmonary function reflect the status of the entire group of participants reliably, it remains that we did not find negative effects of dusty work on pulmonary function in the stone-crushers, who proved to have even better average values than the taxi-drivers. The absence of detectable effects on pulmonary function could be explained by the young age of our participants, as well as by the cross-sectional nature of our study. Indeed, it is generally accepted that the adverse effects of mineral dust exposure on pulmonary function correlate poorly with respiratory symptoms and take many years to become manifest [25,26].

To our knowledge, studies of respiratory health in relation to stone crushing have been exclusively done in quarries using mechanical crushers, and we found no published surveys of workers who only used hand tools to break and crush stones, as was the case in the present study. Only a few longitudinal studies have investigated workers employed in stone quarries. In the USA, a large longitudinal study of granite workers in Vermont showed excessive exposure-related declines in FEV1 and FVC, especially among subjects who failed spirometry or were lost to follow-up [25,26]. In Sweden, Malmberg et al. [27] found that 45 granite crushers (age range 35–77 years) had experienced slightly faster declines in FEV1 after 12 years of work than 45 matched controls. Most published studies of stone crushing workers have been cross-sectional. In Spain, a cross-sectional study of 440 active granite workers by Rego et al. [28] revealed silicosis in 17.5% of subjects, but functional alterations were also found regardless of silicosis (synergistically with smoking). In Singapore, Ng and Chan [29] similarly concluded from a cross-sectional study of 320 workers and ex-workers from two granite quarries that dust exposure was associated with a loss of pulmonary function mainly in the presence of silicosis, but also without silicosis. In Pakistan, Leghari et al. [6] described high prevalences of respiratory symptoms among stone-crushing workers. In India, studies among quartz stone grinders by Tiwari et al. [30,31,32,33] and among sandstone crushers by Singh et al. [34,35] and Rajavel et al. [36] identified silicosis, silico-tuberculosis, and respiratory functional impairment in a high proportion of exposed workers, even at a young age. In Nigeria, Nwibo et al. [37] studied 403 male and female stone quarrying workers [mean age 30 years (SD 9 years)] by means of a questionnaire, spirometry, and chest radiography, and they concluded (without having a control group) that stone quarrying may increase the risk of respiratory symptoms and impaired lung function. In addition, in Nigeria, Isara et al. [38] observed a higher prevalence of various symptoms (mainly chest tightness and cough) and lower levels of FEV1 and FVC among 76 quarry workers [mean 36 years (SD 11years] than among 37 controls. In Libya, Draid et al. [39] found significantly lower spirometric parameters (FEV1, FVC, FEV1/FVC and PEF) among 83 “silica quarry workers” compared to 85 controls. In Ghana, Ahadzi et al. [40] showed, among 524 workers from 30 stone quarries, that self-reported symptoms (eye irritation, breathing difficulties, cough) were inversely related to distance to the main dust source (i.e., crushers) and to the usage of personal protective equipment (worn by around 10% of workers only). We speculate that the levels of dust exposure in our artisanal stone-crushers were lower than those produced when machines are used for crushing stones, but this needs to be evaluated by appropriate environmental measurements.

Moreover, the exposure to dust and the high physical demands associated with stone-crushing may have led to the “healthy worker effect,” whereby people with respiratory impairment tend to quit their job early. Of note, in a study of 272 stone-crushing workers and 123 control agricultural workers in West-Bengal, India, Chattopadhyay et al. [41] found that, contrary to expectation, pulmonary function parameters tended to be better among the exposed group, although there was a higher prevalence of restrictive impairment among the exposed group. As indicated above, in the longitudinal study of pulmonary function among Vermont granite workers, there was evidence of a potent healthy worker effect [26].

On the other hand, we cannot exclude an effect of higher exposure to traffic-related air pollution in the taxi-drivers. These issues can only be investigated by well-powered longitudinal observations.

The strengths of our study include its originality as a first field study of respiratory health among artisanal stone-crushers, with characterization of their metal exposure by urinary biomonitoring. Nevertheless, we also acknowledge several limitations. A first limitation is the cross-sectional design of our survey and the convenience sampling of our population, thus giving rise to uncontrolled selection biases, among which the healthy worker effect is probably an important drawback. We also only included a relatively small group of adult male workers, even though women and children also work on the stone crushing sites. A second limitation is that we do not have information on the levels and composition (including free-silica content) of the dust inhaled by the workers. A further limitation concerns the logistic and technical difficulties of obtaining good quality spirometry in precarious field conditions.

## 5. Conclusions

This study shows that artisanal stone-crushers and taxi-drivers are highly exposed to trace metals in this highly polluted area (Lubumbashi). Wheezing was more prevalent among stone-crushers than among controls and, although no evidence for functional impairment was detected in this preliminary study, this excess wheezing may be indicative of a higher risk of long-term respiratory impairment.

In view of the limitations of our cross-sectional study, which involved small groups of relatively young participants, the lack of a detectable impact on spirometry in the group of stone crushers should not be interpreted as suggesting the absence of risk of respiratory impairment for workers engaged in such artisanal activities. Legislation enforcement and advocacy are warranted to protect the workers.

## Figures and Tables

**Figure 1 ijerph-17-09384-f001:**
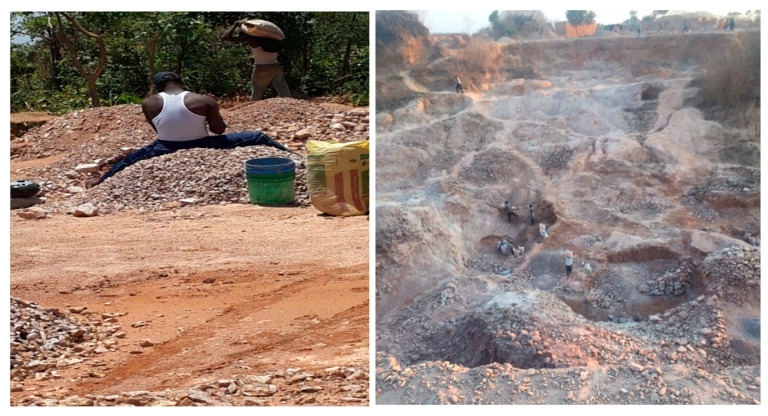
Left panel: An artisanal stone-crusher and a transporter of stones working in precarious conditions. Right panel: General view of the site where stones are manually extracted and then transported on shoulders towards the place where they are kept before being transformed into gravels that will be sold to the purchasers for building roads, bridges, houses.

**Table 1 ijerph-17-09384-t001:** Characteristics of artisanal stone-crushers and taxi drivers in Lubumbashi.

Variables	All (n = 98)	Stone-Crushers (n = 48)	Taxi Drivers (n = 50)	*p*-Value
Age, years	**28 ± 7 [18–50]**	27 ± 5 [18–37]	30 ± 8 [18–50]	0.013
Height, cm	**170 ± 8 [152–194]**	171 ± 8 [152–194]	169 ± 7 [154–185]	0.13
Education ˃12 years	**27 (28%)**	1 (2%)	26 (52%)	0.001
Tobacco smoking	**13 (13%)**	7 (15%)	6 (12%)	0.77
Regular alcohol consumption	**71 (72%)**	36 (75%)	35 (70%)	0.65
Length in work, years	**8 ± 6 [<1–25]**	7 ± 5 [<1–19]	8 ± 7 [<1–25]	0.13
Living close to mining	**17 (17%)**	7 (15%)	10 (20%)	0.60
Exclusive well water consumption	**37 (38%)**	24 (50%)	13 (26%)	0.021

Values are expressed as means ± SD and [range] for continuous variables, and n (%) for categorical variables; *p*-values for the difference between stone-crushers and taxi drivers calculated by *t*-test, Mann–Whitney or Fisher exact test.

**Table 2 ijerph-17-09384-t002:** Urinary concentrations of trace metals/metalloids in the urine of 75 male stone crushers and taxi drivers from Lubumbashi.

Variables	Units	All (n = 75)	Stone-Crushers (n = 41)	Taxi-Drivers (n = 34)		Hoet et al. 2013 (n = 1022)	Banza et al. 2009 (n = 179)
		**GM (95% CI)**	GM (95% CI)	GM (95% CI)	*p*-value	GM (95% CI)	GM 25th–75th percentile
Creatinine	(g/L)	**2.26 (2.05–2.48)**	2.27 (2.00–2.57)	2.24 (1.92–2.61)	0.99	—	—
Li	µg/g creat	**18.8 (16.6–21.4)**	19.5 (16.7–22.7)	18.0 (14.5–22.5)	0.34	22.5 (21.5–23.4)	—
Al	µg/g creat	**12.9 (10.7–15.6)**	12.7 (9.6–16.7)	13.2 (10.2–17.1)	0.50	2.03 (1.94–2.13)	12.4 (6.05–20.7)
Cr	µg/g creat	**0.** **19 (0.** **17–0.** **21)**	0.17 (0.16–0.19)	0.20 (0.17–0.25)	0.26	0.101 (0.10–0.11)	0.17 (0.10–0.24)
Mn	µg/g creat	**0.87 (0.60–1.25)**	1.07 (0.62–1.85)	0.67 (0.42–1.07)	0.39	*	0.32 (0.05–1.87)
Co	µg/g creat	**2.81 (2.18–3.62)**	3.29 (2.39–4.52)	2.33 (1.53–3.54)	**0.049**	0.150 (0.14–0.16)	15.7 (5.27–43.2)
Ni	µg/g creat	**1.86 (1.55–2.22)**	2.40 (1.97–2.93)	1.36 (1.02–1.83)	**0.0007**	1.73 (1.66–1.80)	3.27 (1.90–4.87)
Cu	µg/g creat	**10.4 (9.11–11.8)**	10.1 (8.45–12.0)	10.7 (8.81–13.0)	0.65	6.94 (6.77–7.12)	17.1 (8.44–28.2)
Zn	µg/g creat	**339 (297–388)**	358 (307–416)	319 (250–406)	0.28	229 (220–239)	306 (199–473)
As	µg/g creat	**25.3 (21.0–30.5)**	33.0 (27.1–40.2)	18.3 (13.4–25.1)	**0.006**	15.6 (14.5–16.8)	17.8 (10.9–29.0)
Se	µg/g creat	**18.4 (16.8–20.1)**	19.4 (17.8–21.2)	17.2 (14.5–20.5)	**0.041**	21.3 (21.0–21.7)	12.4 (13.3–20.6)
Mo	µg/g creat	**66.9 (52.5–85.2)**	77.1 (54.6–109)	56.4 (40.0–79.4)	0.49	23.2 (21.5–25.0)	75.2 (46.8–126.7)
Cd	µg/g creat	**0.54 (0.45–0.63)**	0.61 (0.49–0.76)	0.46 (0.36–0.59)	0.065	0.228 (0.216–0.241)	0.75 (0.38–1.16)
Sn	µg/g creat	**0.12 (0.10–0.14)**	0.10 (0.08–0.13)	0.14 (0.11–0.19)	**0.029**	0.286 (0.26–0.31)	0.08 (0.04–0.16)
Sb	µg/g creat	**0.** **05 (0.** **05–0.** **06)**	0.05 (0.04–0.05)	0.06 (0.05–0.08)	**0.005**	0.029 (0.03–0.03)	0.07 (0.04–0.10)
Te	µg/g creat	**0.18 (0.16–0.21)**	0.21 (0.17–0.24)	0.16 (0.13–0.20)	0.12	0.115 (0.11–0.12)	0.09 (0.07–0.12)
Ba	µg/g creat	**1.54 (1.29–1.85)**	1.61 (1.25–2.07)	1.47 (1.11–1.93)	0.32	1.68 (1.59–1.78)	—
Tl	µg/g creat	**0.13 (0.11–0.14)**	0.12 (0.10–0.14)	0.14 (0.12–0.17)	0.087	0.169 (0.16–0.18)	—
Pb	µg/g creat	**2.43 (2.09–2.83)**	2.47 (1.96–3.10)	2.40 (1.95–2.94)	0.70	0.73 (0.70–0.77)	3.17 (1.47–5.49)
U	µg/g creat	**0.** **01 (0.** **01–0.** **02)**	0.02 (0.01–0.02)	0.01 (0.01–0.02)	0.47	*	0.028 (0.013–0.065)

Creat = creatinine, GM = geometric mean (CI = 95% confidence interval), *****: If more than 25 percent of values were below the limit of detection, —: if the value was not available or not shown. Data of Hoet et al. [16] are those reported in their Table 2 for 1022 male and female adults; data of Banza et al. [10] are those reported in their Table 2 for 179 residents (male and female adults and children) living close to mining.

**Table 3 ijerph-17-09384-t003:** Prevalence of reported symptoms (within the past 12 months) among stone-crushers and taxi-drivers.

Variables	All (n = 98)	Stone-Crushers (n = 48)	Taxi-Drivers (n = 50)	*p*-Value
**Oculo-nasal symptoms**				
None	**31 (32%)**	19 (40%)	12 (24%)	0.13
Itchy nose	**13 (13%)**	6 (13%)	7 (14%)	1.00
Itchy eyes	**7 (7%)**	2 (4%)	5 (10%)	0.44
Runny nose	**13 (13%)**	5 (10%)	8 (16%)	0.55
Sneezing	**22 (22%)**	11 (23%)	11 (22%)	1.00
Stuffy nose	**12 (12%)**	5 (10%)	7 (14%)	0.76
**Respiratory Symptoms**				
None	**44 (45%)**	17 (35%)	27 (54%)	0.072
Wheezing	**14 (14%)**	11 (23%)	3 (6%)	**0.021**
Cough	**14 (14%)**	6 (13%)	8 (16%)	0.78
Phlegm	**16 (16%)**	10 (21%)	6 (12%)	0.28
Shortness of breath	**10 (10%)**	4 (8%)	6 (12%)	0.74

Values are presented as numbers and percentages; *p*-values for the difference between stone-crushers and taxi drivers calculated by the Fisher exact test.

**Table 4 ijerph-17-09384-t004:** Unadjusted and adjusted associations between independent variables and respiratory symptoms in the overall population.

Variables	Group (Stone-Crushers vs. Taxi-Drivers)		Age		Tobacco Smoking (Yes vs. No)		Living Close to Mining (Yes vs. No)	
	uOR [95% CI]	aOR [95% CI]	uOR [95% CI]	aOR [95% CI]	uOR [95% CI]	aOR [95% CI]	uOR [95% CI]	aOR [95% CI]
Itchy nose	0.87 [0.27–2.83]	0.85 [0.25–2.83]	1.23 [0.75–1.69]	1.02 [0.92–1.12]	0.6 [0.18–2.04]	0.50 [0.06–4.34]	1.32 [0.40–5.02]	1.49 [0.35–6.19]
Itchy eyes	0.39 [0.07–2.12]	0.36 [0.06–2.03]	0.88 [0.67–1.56]	1.06 [0.92–1.23]	1.82 [0.54–4.88]	2.91 [0.45–18.52]	0.87 [0.19–4.09]	2.35 [0.37–14.73]
Runny nose	0.61 [0.18–2.01]	0.50 [0.14–1.72]	0.99 [0.71–1.32]	1.10 [0.99–1.24]	0.87 [0.44–2.27]	0.42 [0.04–3.75]	0.70 [0.19–2.48]	1.66 [0.38–7.29]
Sneezing	1.05 [0.41–2.72]	0.94 [0.35–2.50]	1.21 [0.88–1.52]	1.03 [0.95–1.12]	0.93 [0.23–3.81]	1.21 [0.19–7.62]	**1.61 [1.22–2.28]**	0.73 [0.18–2.87]
Stuffy nose	0.71 [0.21–2.43]	0.59 [0.16–2.12]	1.15 [0.81–1.67]	1.03 [0.93–1.13]	**2.32 [1.19–5.51]**	1.14 [0.21–6.19]	NC	NC
Wheezing	**4.66 [1.21–17.9]**	**4.45 [1.09–18.2]**	**0.90 [0.81–0.99]**	**1.17 [1.02–1.35]**	0.80 [0.20–3.81]	1.10 [0.19–6.37]	2.09 [0.45–11.6]	2.37 [0.48–11.55]
Cough	0.75 [0.23–2.34]	0.63 [0.18–2.06]	1.02 [0.98–1.13]	**1.14 [1.01–1.29]**	1.81 [0.72–4.60]	1.78 [0.39–8.02]	1.83 [0.40–8.19]	2.54 [0.47–13.10]
Phlegm	1.92 [0.64–5.80]	1.76 [0.57–5.41]	1.13 [0.90–1.62]	1.05 [0.95–1.15]	1.32 [0.56–3.62]	1.57 [0.36–6.67]	0.69 [0.20–3.29]	1.38 [0.33–5.76]
SOB	0.66 [0.17–2.52]	0.56 [0.13–2.44]	**1.05 [1.01–1.13]**	**1.28 [1.06–1.55]**	**2.48 [1.09–5.94]**	3.59 [0.67–19.14]	0.51 [0.10–2.19]	4.84 [0.83–28.32]

uOR: Unadjusted odds ratio, aOR: Adjusted odds ratio, adjusted for a group of workers, age, tobacco smoking, and living close to mining (i.e., within sight), NC: Not calculated; statistically significant associations are shown in bold.

**Table 5 ijerph-17-09384-t005:** Pulmonary Function Tests in stone-crushers and taxi-drivers.

Variables	All (n = 21)	Stone-Crushers (n = 10)	Taxi-Drivers (n = 11)	*p*-Value
FVC (% predicted)	**98.5 ± 17.8 [68.0–127.0]**	105.4 ± 15.9 [81.0–127.0]	92.2 ± 17.8 [68.0–125.0]	0.048
FEV1 (% predicted)	**95.8 ± 18.7 [50.0–124.0]**	104.4 ± 13.7 [84.0–124.0]	88.0 ± 19.6 [50.0–124.0]	0.052
FEV1/FVC (%)	**84.2 ± 8.5 [58.4–96.6]**	86.5 ± 6.9 [73.6–96.6]	82.2 ± 9.6 [58.4–94.4]	0.31
PEF (% predicted)	**95.2 ± 22.2 [56.0–138.0]**	101.5 ± 21.1 [69.0–135.0]	89.5 ± 22.46 [56.0–138.0]	0.16
MEF25–75 (% predicted)	**71.3 ± 23.3 [15.0–123.0]**	81.0 ± 18.9 [55.0–123.0]	62.5 ± 24.2 [15.0–94.0]	0.16
MEF50 (% predicted)	**80.4 ± 25.6 [18.0–125.0]**	87.8 ± 18.1 [66.0–125.0]	73.7 ± 30.2 [18.0–117.0]	0.16
MEF25 (% predicted)	**66.8 ± 28.1 [7.0–108.0]**	79.1 ± 20.7 [51.0–108.0]	55.7 ± 30.1 [7.0–102.0]	0.078

FVC (Forced Vital Capacity), FEV1 (Forced Expiratory Volume in 1 Second), PEF (Peak Expiratory Flow), MEF (Maximal Expiratory Flow at given percentage of FVC), all expressed as percent predicted: values are means ± standard deviation [range]; *p*-values calculated by the Mann–Whitney test.

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
