# Peer review of "Respiratory Health and Urinary Trace Metals among Artisanal Stone-Crushers: A Cross-Sectional Study in Lubumbashi, DR Congo"

_ijerph, 2020, doi:10.3390/ijerph17249384_

Round 1
Reviewer 1 Report
The subject under study is not new. There are multiple studies about the respiratory health of stoneworkers.
The objective of the study is correctly indicated. However, no hypothesis is raised.
Regarding the methodology, it is worth highlighting the low number of participants, as well as obtained valid spirometries, and the absence of a chest X-ray. The cases and the controls have not been paired, so they present different characteristics, and are not comparable.
The study has been approved by the Ethics Committee of the center, however the participants have not signed the document of participation in it.
Results are expressed clearly.
The authors misinterpret the results. Despite the multiple biases and limitations of the study, they conclude that there are no respiratory alterations among stoneworkers. There is multiple literature that contradicts this conclusion.
For all these reasons, I propose to reject the article.
Author Response
Thank you for this comment.
- We agree that there have been many studies in stoneworkers. However, the novelty of our design is that this is the first study combining urinary biomonitoring to assess metal exposure, and spirometry to assess pulmonary function in Africa, especially in one of the most polluted area in the world (African Copperbelt) where diggers and stones crushers are exposed to dust. To the best of our knowledge, this is the first study to be done in this highly polluted region.
- The hypothesis of our cross-sectional study was that the stone crushers would differ from the control subjects.
- The controls (taxi-drivers) and stone-crushers were not paired, but there was no need to do so. This was not a case-control study, where pairing may be advisable, but a cross-sectional study where group comparisons are perfectly acceptable, provided the groups come from the same source population (which was the case). We agree (as stated in the limitations) that the convenience sampling and the low numbers of participants are limitations.
- The absence of X-rays is also a limitation due to logistic and financial constrains in this medically underserved setting in one of the poorest places in the world. However, it is doubtful that substantial evidence of pneumoconiosis would have been detected with chest x-rays in our young population, given the type of work done and the other findings.
- Our committee for medical ethics agrees with obtaining oral informed consent in observational studies without invasive procedures or therapeutic trials.
Concerning the interpretation of our results, we did not conclude that there were no respiratory alterations in the stone-crushers. We established that they reported more wheezing but this was not (yet) reflected in spirometric alterations, possibly because we had too few valid measurements, but mainly because our population was still relatively young and possibly had relatively low dust exposure (compared to workers exposed to mechanical crushers) and because of the healthy worker effect; moreover, our control group of taxi-drivers was not free from exposure to inhaled traffic-related pollutants. These caveats were clearly expressed in the discussion and we have now also expanded the review of the literature on respiratory effects in stone workers.
Reviewer 2 Report
The manuscript describes a cross-sectional study in which the exposure levels to metals of Artisanal Stone-crushers are compared with a control sample of taxi-drivers.
The study is original; the subject is very interesting; it is well designed and meets the objectives.
As for weaknesses, the systematical use of sentences in the first person is pointed out. This use is not good practice in scientific literature and weakens the work strongly. This aspect must be corrected throughout the manuscript.
Introduction chapter should also be reinforced to contextualise better the work, as well as the motivation of the study.
The number of references is also very small (19), eight of which belonging to the group that developed the present work. Although this last fact demonstrates continued work in the area, there appears to be a lack of knowledge or interest in the work of other authors, which must be corrected.

Author Response
- Thank you very much for these suggestions. We respectfully disagree with the criticism regarding the use of the first person; most specialists now recommend writing scientific articles in the active form rather than in the passive form.
- Introduction chapter should also be reinforced to contextualize better the work, as well as the motivation of the study, we are grateful for this comment, and have expanded the introduction accordingly in the revised version
- The number of references is also very small (19); we agree that the number references is low, but this was due to the scarcity of publications on pulmonary health among artisanal workers involved in stone crushing. We have now included (in the introduction and the discussion) a large number of references to the literature about dust exposure and health effects related to stone-crushing,(however, all of them related to crushing by machines rather than by hand tools, as in the present study.
Reviewer 3 Report
The paper “Respiratory Health and Urinary Trace Metals among Artisanal Stone-crushers: a Cross-sectional Study in Lubumbashi, DR Congo by Kayembe-Kitenge et al., is a cross sectional study that compares the urinary excretion of metals, respiratory function, and self reported respiratory and upper airway symptoms in stone-crushers and taxi drivers. This study used a straight-forward and well executed methodology although admittedly, there were difficulties in obtaining valid spirometry results. This paper provides valuable insight into the exposures sustained by artisanal workers under what appear to be challenging conditions.
My comments are as follows:
- In the Introduction or Methods, elaborate on the stone-crushing activities that lead to the exposure. How is the stone crushed? Is any equipment used? What other activities are performed e.g. transport of the crushed stone. Figures A and B address the work being done but are not mentioned in the text. Figure A is distorted and not very informative.
- Table 2 has a lot of good information but is difficult to review. Have you considered converting this to a figure?
- The papers by Hoet and Banza are used as reference groups which is a good idea. In the table, it would be more informative to refer to the Hoet paper as being a Belgian comparison, and the Banza paper as being a regional comparison. Also, the methods should include a short description of these papers.
Author Response
Thank you for this comment. Our answers are as follows:
- Stones are crushed manually, using the stone versus stones or a hammer, people transport themselves the stones on their shoulders from the ground to the work place. The legend of the figure A and B address the work being done, and we wrote a sentence concerning these photographs in the introduction (see line 54).
- To convert the table2 to a figure seems very difficult to convey the message that we would like to show to our audience.
- Thank you for the suggestion to specify the populations covered in the Hoet et al and Banza et al articles. The same analytical methodology was used in both papers and a few more methodological details have been provided in the methods section and the studied populations have been described in the results section.
Round 2
Reviewer 1 Report
Few changes have been made (mostly deleting paragraphs and rewriting in better english).
It is still a poor constructed study. There is a low number of participants, and very few valid spirometries. The cases and the controls present different characteristics.
It still has multiple biases and limitations.
Author Response
Thank you for the comment,
We hereby respond to your questions:
- We do not agree that our study lacks novelty because this is the first study combining urinary biomonitoring to assess metal exposure, and spirometry to assess pulmonary function in Africa, and we found no published surveys of workers who only used hand tools to break and crush stones especially in one of the most polluted area in the world (African Copperbelt) where diggers and stones crushers are exposed to dust. To the best of our knowledge, this is the first study to be done in this highly polluted region;
- We had made substantial additions to our manuscript to compare our findings with those of others (22 references added with a detailed description of their findings in a specific paragraph in the discussion) and;
- We did acknowledge in our discussion that the study was not perfect and we had carefully addressed its limitations. Nevertheless, we have added the following cautionary sentence to the conclusion: “In view of the limitations of our cross-sectional study, which involved small groups of relatively young participants, the lack of a detectable impact on spirometry in the group of stone crushers should not be interpreted as suggesting no risk of respiratory impairment for workers engaged in such artisanal activities.”